# Insulin-like growth factor 1 receptor expression correlates with programmed death ligand 1 expression and poor survival in non-small cell lung cancer

Hiroaki Nagamine[1,2]☯, Masakazu Yashiro[3]☯*, Megumi Mizutani[2], Akira Sugimoto[2], Yoshiya Matsumoto[2], Yoko Tani[4], Kenji Sawa[4], Hiroyasu Kaneda[4], Kazuhiro Yamada[2], Tetsuya Watanabe[2], Kazuhisa Asai[2], Satoshi Suzuki[5‡], Tomoya Kawaguchi[2,4]

1 Department of Respiratory Medicine, Graduate School of Medicine, Osaka City University, Osaka, Japan, 2 Department of Respiratory Medicine, Graduate School of Medicine, Osaka Metropolitan University, Osaka, Japan, 3 Molecular Oncology and Therapeutics, Graduate School of Medicine, Osaka Metropolitan University, Osaka, Japan, 4 Department of Clinical Oncology, Graduate School of Medicine, Osaka Metropolitan University, Osaka, Japan, 5 Department of Thoracic Surgery, Graduate School of Medicine, Osaka Metropolitan University, Osaka, Japan

☯ These authors contributed equally to this work.
‡ SS also contributed equally to this work.
* i21496f@omu.ac.jp

**Data Availability Statement:** All relevant data are within the manuscript and its Supporting information files.

## Abstract

The insulin-like growth factor 1 receptor (IGF1R) has been associated with growth and metastasis in various cancers. However, its role in postoperative recurrence and prognosis in lung cancer lacks clear consensus. Therefore, this study aimed to investigate the potential relationship between IGF1R and postoperative recurrence as well as long-term survival in a large cohort. Additionally, we assessed the relationship between IGF1R and programmed death ligand 1 (PD-L1) expression. Our study encompassed 782 patients with non-small cell lung cancer (NSCLC). Immunostaining of surgical specimens was performed to evaluate IGF1R and PD-L1 expression. Among the patients, 279 (35.8%) showed positive IGF1R expression, with significantly worse relapse-free survival (RFS) and overall survival (OS). Notably, no significant differences in RFS and OS were observed between IGF1R-positive and -negative groups in stages 2 and 3. However, in the early stages (0–1), the positive group displayed significantly worse RFS and OS. In addition, PD-L1 expression was detected in 100 (12.8%) patients, with a significant predominance in the IGF1R-positive. IGF1R may serve as a prognostic indicator and a guide for perioperative treatment strategies in early-stage lung cancer. In conclusion, our findings underscore an association between IGF1R expression and poor survival and PD-L1 expression in NSCLC.

## Introduction

The insulin-like growth factor 1 receptor (IGF1R) is expressed in various cancers [1–3] and is associated with cancer growth [4], metastasis [5, 6], and malignancy [7, 8]. In lung cancer,

**Funding:** The author(s) received no specific funding for this work.

**Competing interests:** The authors have declared that no competing interests exist.

particularly non-small cell lung cancer (NSCLC), IGF1R expression is associated with carcinogenesis and proliferation [9, 10]. Nevertheless, the association between IGF1R expression and postoperative prognosis as well as recurrence in lung cancer has been inconsistent [11–14], lacking a definitive consensus.

In recent years, the expression of programmed death ligand 1 (PD-L1) has attracted attention as a predictor of immune checkpoint inhibitor (ICI) efficacy in the treatment of lung cancer [15]. Although the expression of IGF1R and PD-L1 has been implicated in head and neck squamous cell carcinoma [16], their role in lung cancer remains unknown. The development of lung cancers, particularly squamous cell carcinoma (SCC), has been associated with smoking [17]. Moreover, smoking has been known to be involved in tumor-induced IGF1R expression [10]. However, it remains unclear whether IGF1R expression is associated with smoking and PD-L1 expression in actual clinical practice and whether it is involved in postoperative recurrence and survival in lung cancer. Therefore, this study aimed to investigate the potential associations between IGF1R expression, PD-L1 expression, smoking history, postoperative recurrence, and survival in lung cancer. To our knowledge, this is the first study to comprehensively evaluate the relationship between IGF1R and these critical factors using long-term observations on a large scale. We anticipate that our findings will contribute to the ongoing efforts to optimize lung cancer management, leading to more effective and personalized therapeutic interventions in the realm of precision medicine.

## Materials and methods

### Patient and data collection

A total of 782 patients with NSCLC who underwent surgery at the Osaka Metropolitan University Hospital between January 2010 and December 2019 were included in this study. Cancer stage for all patients was assessed according to the Union for International Cancer Control, 8th edition. Relapse-free survival (RFS) was calculated from the date of surgery to the date of first recurrence or death from any cause. Overall survival (OS) was calculated from the date of surgery to the date of death from any cause. Our observations were conducted retrospectively, with August 31, 2023, as the data cutoff date, spanning a maximum of 10 years. At the cut-off date or 10 years after the start of observation, cases with no events were terminated from observation. Between September 1, 2023, and October 31, 2023, we accessed medical records for the collection of information, including personally identifiable details, related to registered patients. This study was approved by the Osaka City University Ethics Committee (reference number 2019–006). Informed consent was obtained from each patient. This study was conducted according to the principles of the Declaration of Helsinki.

### Immunohistochemistry

Immunohistochemical staining was performed on paraffin-embedded sections of primary surgically resected specimens obtained from patients with NSCLC. Slides were deparaffinized using xylene and rehydrated with gradually decreasing concentrations of ethyl alcohol. The slides were then incubated with methanol containing 3% hydrogen peroxide at room temperature for 15 min to eliminate endogenous peroxidase activity. Subsequently, the slides were heated in an autoclave at 105°C for 10 min in Target Retrieval Solution (Dako, Santa Clara, CA, USA). Following this, the slides were blocked with 10% normal mouse serum for 10 min at room temperature and incubated overnight at 4°C with the anti-IGF1R antibody (NB110-87052, 1:200; Novus Biologicals LLC, Centennial, CO, USA). Subsequently, the samples were incubated with a biotinylated secondary antibody for 10 min at room temperature, followed by treatment with streptavidin-peroxidase reagent for 5 min at room temperature, incubation

in diaminobenzidine for 2 min at room temperature, and counterstaining with 100% Mayer's hematoxylin for 40 s at room temperature. For PD-L1 staining, sections were blocked with 10% normal rabbit serum, and an anti-PD-L1 antibody (ab205921, 1:150, Abcam, Cambridge, UK) was used as the primary antibody, followed by reaction with diaminobenzidine for 7 min.

## Statistical analysis

The $\chi 2$ test was employed to evaluate the significance of differences in patient characteristics between the IGF1R-positive and IGF1R-negative groups. Multiple logistic regression analysis was utilized to analyze the relationship between IGF1R and PD-L1 expression, incorporating age, sex, smoking history, Eastern Cooperative Oncology Group Performance Status (ECOG PS), histology, pStage, pleural invasion, lymphatic invasion, and vascular invasion as factors. The Mann–Whitney U test evaluated differences in smoking index or maximum tumor diameter between the IGF1R-positive and negative groups. Survival curves were generated using Kaplan–Meier method, and the log-rank test was used to compare cumulative survival durations in the patient groups. In addition, the Cox proportional hazards model was used to compute univariable and multivariable hazard ratios for the study parameters. Analyses were performed using EZR in R Commander version 1.54 (Saitama Medical Center, Jichi Medical University, Saitama, Japan). In all tests, $p < 0.05$ was considered to indicate statistical significance.

## Results

### Patient characteristics and the association of IGF1R expression with smoking history and tumor size

A total of 782 patients were included in this study. Both IGF1R and PD-L1 were predominantly stained in the cytoplasm (Fig 1A–1C). IGF1R was identified as positive in 279 patients (35.8%). The median age of the participants was 70.0 years (range = 33–91), with 497 (63.6%) patients being male. Five hundred twenty-one patients had adenocarcinoma, 217 had SCC, 14 had adenosquamous cell carcinoma, 11 had large cell carcinoma, nine had pleomorphic carcinoma, three had large cell neuroendocrine carcinoma, two had adenocarcinoma in situ, one had mixed adenocarcinoma and small cell carcinoma, one had SCC and small cell carcinoma, one had NSCLC not otherwise specified, one had mucoepidermoid carcinoma, and one had atypical adenomatous hyperplasia. The IGF1R-positive group exhibited significantly higher proportions of patients who were $\geq$65 years old, male, had a smoking history, a performance status (PS) $\geq$1, had SCC, pStage$\geq$2, exhibited pleural invasion, had vascular invasion, or showed PD-L1 expression (all p<0.01, p<0.01, p<0.01, p<0.01, p<0.01, p<0.01, p = 0.02, p<0.01, p<0.01, respectively) (Table 1). Multivariable analysis revealed that PD-L1 expression is associated with IGF1R expression (Table 2). Both IGF1R-positive and PD-L1-positive group had a significantly higher smoking index (p<0.01) (Fig 2A and 2B), and the IGF1R-positive group had a significantly larger maximum tumor diameter (p<0.01) (Fig 2C).

### Relationship between IGF1R and relapse-free survival or overall survival

In all patients, IGF1R positivity was associated with worse RFS (p<0.01) and OS (p<0.01) (Fig 3A and 3B). When evaluated by stage, IGF1R positivity was associated with worse RFS (p<0.01) and OS (p<0.01) in Stage 0–1 (Fig 4A and 4B). However, no significant differences in RFS or OS were observed between IGF1R-positive and negative patients in Stage 2 (p = 0.60 and p = 0.80, respectively) (Fig 5A and 5B) or Stage 3 (p = 0.89 and p = 0.15, respectively) (Fig 6A and 6B). In univariable analysis of RFS in stage 0–1 patients, IGF1R positivity, male,

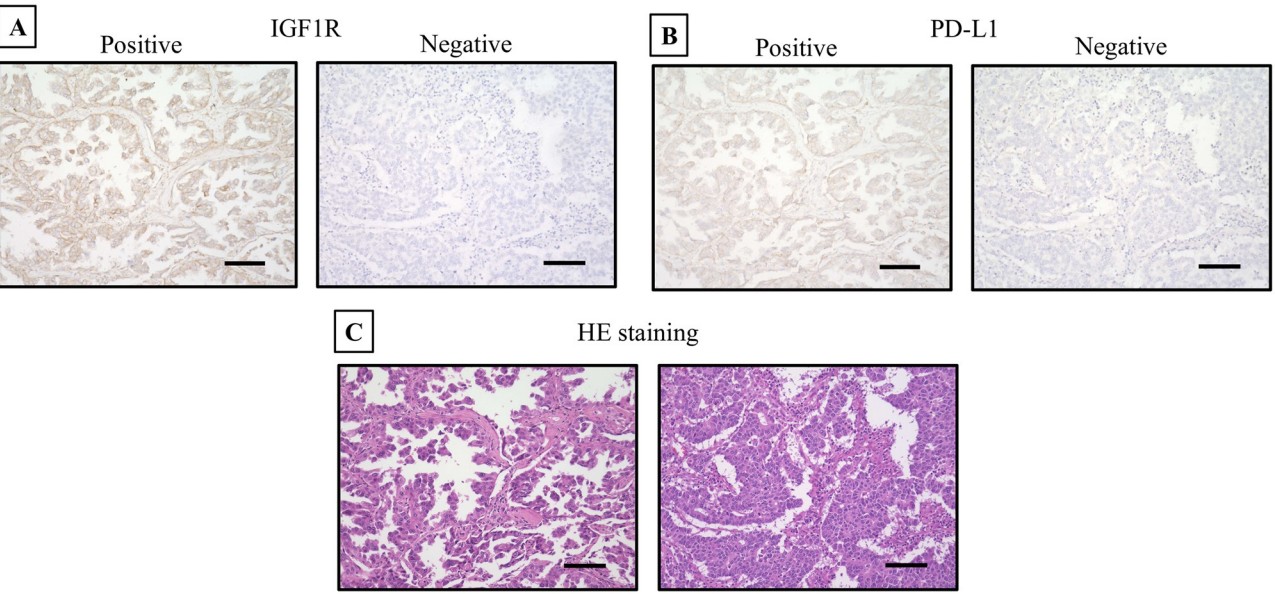

**Fig 1. Immunostaining for IGF1R and PD-L1, and hematoxylin-eosin staining. (A)** Immunostaining for IGF1R reveals predominant cytoplasmic staining. **(B)** Immunostaining for PD-L1 reveals predominant cytoplasmic staining. **(C)** Hematoxylin-eosin staining.

smoking history, ECOG PS ≥1, pStage 2–3, pleural invasion, lymphatic invasion, and vascular invasion were all associated with significantly worse RFS (Table 3). In the multivariable analysis of RFS in stage 0–1 patients, smoking history, ECOG PS≥1, lymphatic invasion, and vascular invasion were associated with significantly worse RFS (Table 3). In univariable analysis of OS in stage 0–1 patients, IGF1R positivity, male, smoking history, ECOG PS≥1, SCC and vascular invasion were all associated with significantly worse OS (Table 4). In the multivariable analysis of OS in stage 0–1 patients, ECOG PS≥1 was associated with significantly worse OS (Table 4).

## Discussion

Although previous reports have not consistently linked IGF1R expression to postoperative recurrence or prognosis in lung cancer [11–14], our large-scale, long-term observational study demonstrates that IGF1R expression is associated with worse RFS and OS, particularly in the early stages. In addition, in this study, IGF1R expression was significantly correlated with PD-L1 expression, smoking history, and tumor size.

The IGF1R-positive group also showed significantly larger tumor sizes. IGF1R is a factor associated with tumor growth [4]. Upon binding with its ligands IGF1 or IGF2, IGF1R activates downstream signaling pathways such as the phosphoinositide 3-kinase (PI3K)/Akt and extracellular signal-regulated kinase (ERK) pathways, which leads to tumor cell proliferation [18–20]. This is consistent with the result of the relationship between IGF1R expression and tumor size. In addition, IGF1R expression was associated with shorter RFS and OS, particularly in early-stage lung cancer. It has been known that stage 1 has a lower frequency and amount of postoperative molecular residual disease (MRD) compared to stage 2 or 3 in lung cancer [21]. Although the immune system may eliminate minimal MRD in early-stage lung cancer, our results suggested that IGF1R expression was associated with PD-L1 expression, which might have allowed cancer cells to escape immune surveillance [22, 23] and increased

**Table 1. Patient characteristics.**

| | No. of all patients | IGF1R positive | IGF1R negative | p-value |
| --- | --- | --- | --- | --- |
| | n = 782 | n = 279 (35.8%) | n = 503 (64.2%) | |
| Age | | | | |
| <65 | 200 (25.6%) | 50 (17.9%) | 150 (29.8%) | <0.01 |
| ≥65 | 582 (74.4%) | 229 (82.1%) | 353 (70.2%) | |
| Sex | | | | |
| Male | 497 (63.6%) | 210 (75.3%) | 287 (57.1%) | <0.01 |
| Female | 285 (36.4%) | 69 (24.7%) | 216 (42.9%) | |
| Smoking history | | | | |
| Current/Former | 587 (75.1%) | 243 (87.1%) | 344 (68.4%) | <0.01 |
| Never | 195 (24.9%) | 36 (12.9%) | 159 (31.6%) | |
| ECOG PS | | | | |
| 0 | 573 (73.3%) | 193 (68.8%) | 380 (75.4%) | <0.01 |
| 1–3 | 115 (14.7%) | 56 (20.2%) | 59 (11.7%) | |
| Histology | | | | |
| Sq | 218 (27.9%) | 137 (49.1%) | 81 (16.1%) | <0.01 |
| Non Sq | 564 (72.1%) | 142 (50.9%) | 422 (83.9%) | |
| pStage | | | | |
| 0–1 | 494 (63.2%) | 137 (49.1%) | 357 (71.0%) | <0.01 |
| 2–3 | 288 (36.8%) | 142 (50.9%) | 146 (29.0%) | |
| Adjuvant therapy | | | | |
| Done | 192 (24.6%) | 72 (25.8%) | 120 (23.9%) | 0.55 |
| None | 590 (75.4%) | 207 (74.2%) | 383 (76.1%) | |
| Pleural invasion | | | | |
| Positive | 228 (29.2%) | 96 (34.4%) | 132 (26.2%) | 0.02 |
| Negative | 548 (70.1%) | 181 (64.9%) | 367 (73.0%) | |
| Lymphatic invasion | | | | |
| Positive | 212 (27.1%) | 78 (28.0%) | 134 (26.6%) | 0.74 |
| Negative | 566 (72.4%) | 200 (71.7%) | 366 (72.8%) | |
| Vascular invasion | | | | |
| Positive | 169 (21.6%) | 98 (35.1%) | 71 (14.1%) | <0.01 |
| Negative | 609 (77.9%) | 180 (64.5%) | 429 (85.3%) | |
| PD-L1 | | | | |
| Positive | 100 (12.8%) | 68 (24.4%) | 32 (6.4%) | <0.01 |
| Negative | 682 (87.2%) | 211 (75.6%) | 471 (93.6%) | |

IGF1R, insulin-like growth factor 1 receptor; ECOG PS, Eastern Cooperative Oncology Group Performance Status; Sq, squamous cell carcinoma; PD-L1, programmed death ligand 1.

the likelihood of relapse in IGF1R-positive patients. Additionally, IGF1R expression may

**Table 2. Multivariate analysis of the relationship between IGF1R expression and PD-L1 expression.**

| Factor | Reference | OR (95%CI) | p-value |
| --- | --- | --- | --- |
| PD-L1 | Negative | 3.35 (1.95–5.73) | <0.01 |

IGF1R, insulin-like growth factor 1 receptor; OR, odds ratio; CI, confidence intervals; PD-L1, programmed death ligand 1.

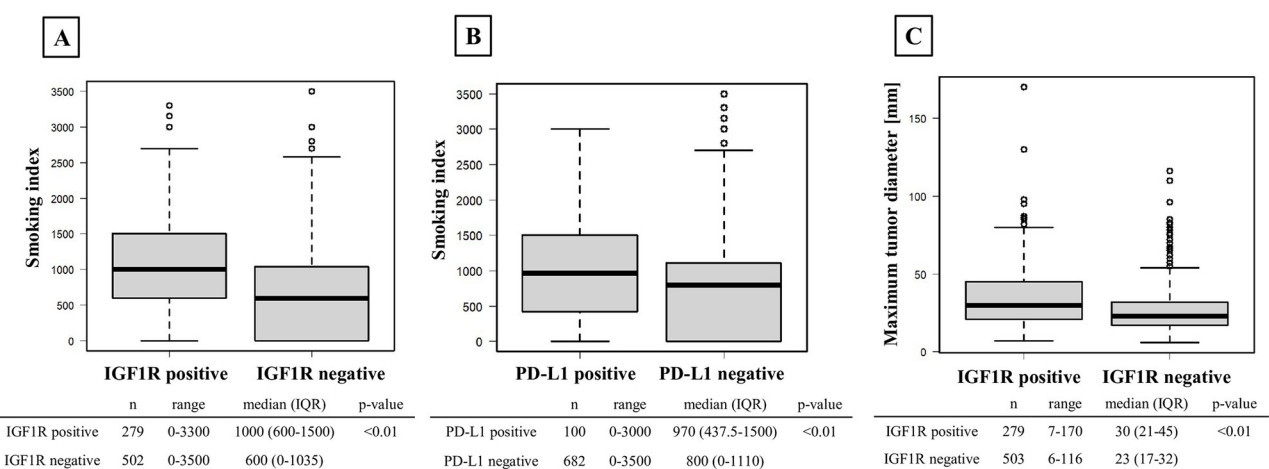

**Fig 2. The association between IGF1R or PD-L1 and smoking index, as well as the correlation between IGF1R and tumor size. (A)** Comparison of smoking index between the IGF1R-positive and negative groups. The IGF1R-positive group had a significantly larger smoking index. **(B)** Comparison of smoking index between PD-L1-positive and negative groups. The PD-L1-positive group had a significantly larger smoking index. **(C)** Comparison of maximum tumor diameter between IGF1R-positive and negative groups. The IGF1R-positive group had a significantly larger maximum tumor diameter.

facilitate tumor growth, leading to earlier recurrence compared to IGF1R-negative cases. Also, IGF1R expression has been known to correlate with the expression of ATP–binding cassette subfamily G member 2 (ABCG2) [24], a potential marker for cancer stem cells [25] and involved in chemoresistance [26]. These findings might explain the treatment resistance of

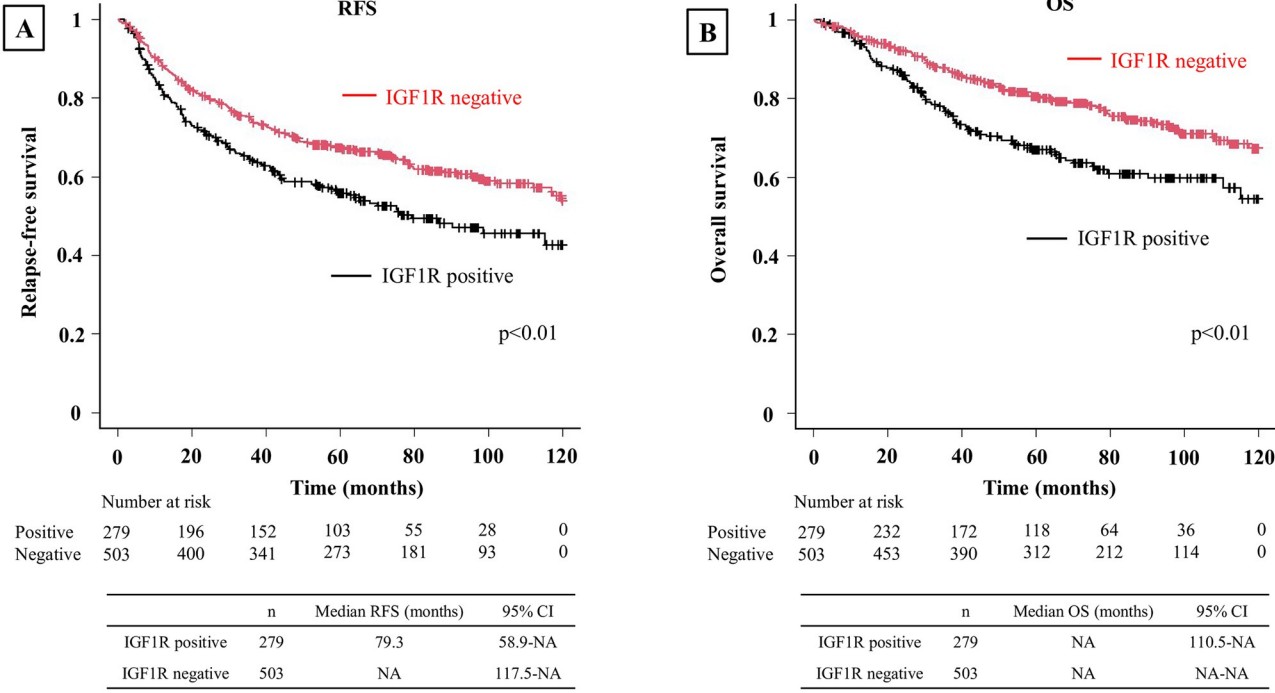

**Fig 3. Kaplan-Meier curve depicting RFS and OS for all patients. (A)** Kaplan-Meier curve depicting RFS for all patients. The IGF1R-positive group exhibited significantly worse RFS. **(B)** Kaplan-Meier curve depicting OS for all patients. The IGF1R-positive group exhibited significantly worse OS.

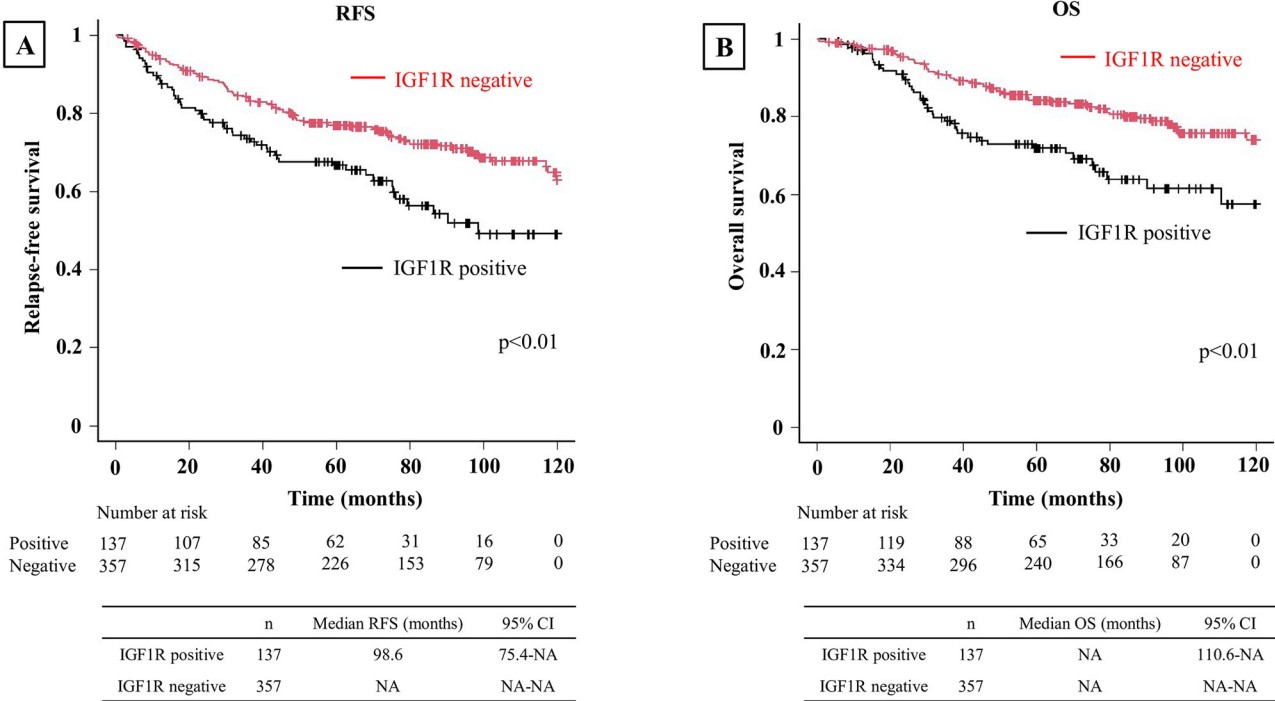

**Fig 4. Kaplan-Meier curve depicting RFS and OS in stage 0–1 patients.** (A) Kaplan-Meier curve depicting RFS in stage 0–1 patients. The IGF1R-positive group exhibited significantly worse RFS. (B) Kaplan-Meier curve depicting OS in stage 0–1 patients. The IGF1R-positive group exhibited significantly worse OS.

recurrent tumors and may account for the worse OS observed in early-stage lung cancer with IGF1R positivity, where recurrence is more frequent. Many treatment strategies using ICI for neoadjuvant and adjuvant chemotherapy have been developed in recent years [27, 28]. However, due to the favorable prognosis of early-stage lung cancer, it is less amenable to the benefits of neoadjuvant and adjuvant chemotherapy. Based on our study findings, IGF1R expression may serve as an informative marker for determining appropriate perioperative chemotherapy for patients with early-stage lung cancer. Furthermore, despite promising preclinical indications, IGF1R inhibitors have not proven consistently effective in various types of cancer in clinical trials [29–31]. However, recent experiments in mice have demonstrated that the combination of an IGF1R inhibitor and an anti-PD-1 antibody synergistically inhibits tumor growth [32, 33]. IGF1R expression is correlated with the tumor microenvironment and inhibits the activation of effector cytotoxic CD8[+] T cells [34, 35]. Therefore, inhibition of the IGF1R pathway has the potential to enhance the effectiveness of ICI by activating effector cytotoxic CD8 T cells [36]. From those results, Combining IGF1R inhibitors with ICI may hold promise for future therapies, especially as perioperative treatment for early-stage lung cancer.

In this study, we observed a positive correlation between the expression of IGF1R and PD-L1. Notably, in head and neck squamous cell carcinoma, where smoking poses a risk [37], Interleukin-6 (IL-6) has been reported to upregulate IGF1R and PD-L1 expression [16]. In the present study, the IGF1R-positive group had a significantly higher smoking index, consistent with previous studies linking IGF1R expression to smoking [14]. Smoking induces IL-6 expression in the lungs [38], potentially contributing to the upregulation of both IGF1R and PD-L1 expression in lung cancer. Additionally, the IGF1R-positive group had a significantly

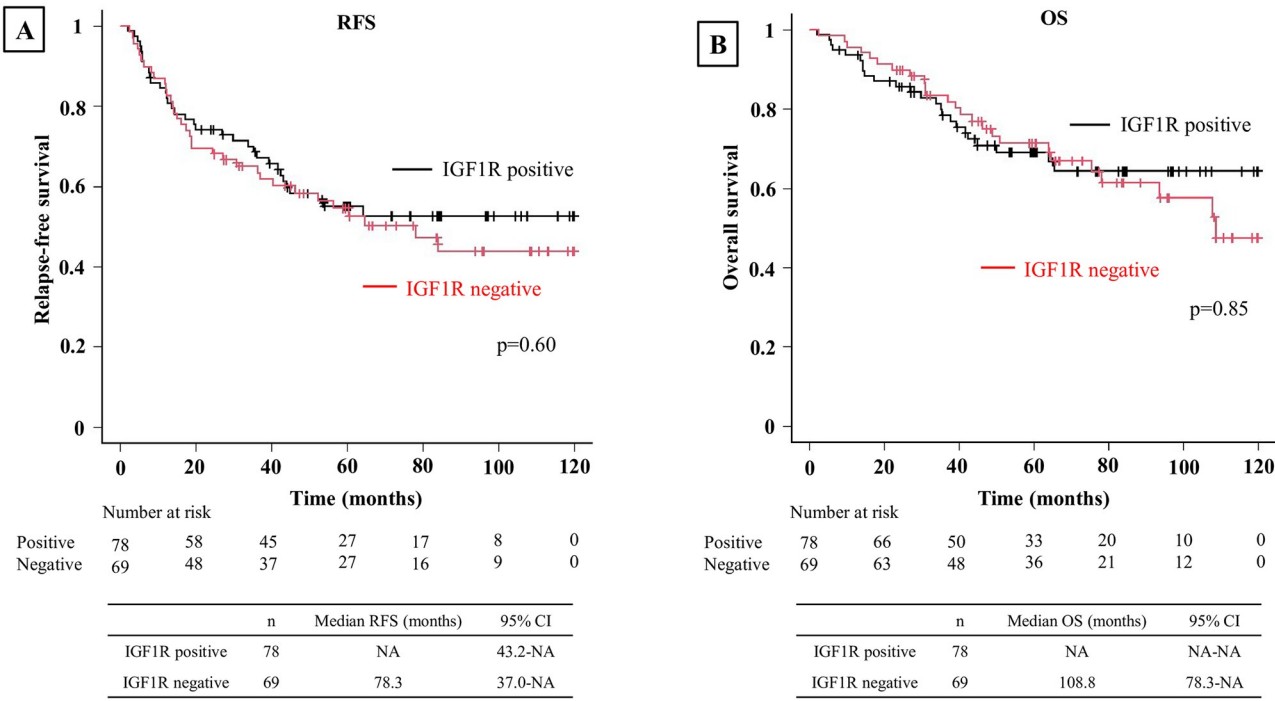

**Fig 5. Kaplan-Meier curve depicting RFS and OS in stage 2 patients. (A)** Kaplan-Meier curve depicting RFS in stage 2 patients. No significant difference was observed between the IGF1R-positive and negative groups. **(B)** Kaplan-Meier curve for OS in stage 2 patients. No significant difference was observed between the IGF1R-positive and negative groups.

higher proportion of SCC, widely known to be associated with smoking [17]. Smoking-induced inflammation also enhances COX2 expression, which is associated with IGF1R expression in lung cancer [10]. Various smoking-related mechanisms, such as IL-6 and COX-2, may elucidate the association between IGF1R expression, smoking, SCC, and PD-L1 expression. However, further research is needed to elucidate the mechanism of co-expression of IGF1R and PD-L1. Our unpublished data indicate that OMUL-1, a human squamous lung cancer cell line, expresses high levels of both IGF1R and PD-L1. This cell line may be useful for analyzing the mechanisms responsible for the co-expression of IGF1R and PD-L1.

IGF1R expression was correlated with various factors, such as PD-L1, sex, and age, which differed between the IGF1R-positive and negative groups. Additionally, IGF1R expression was positively correlated with smoking index. Given the high prevalence of male smokers in Japan [39], the prevalence of IGF1R-positive lung cancer may be higher in males. Moreover, the smoking index tends to be higher in older patients, and the population of nonsmokers has decreased in recent years [39], which may explain the large number of older adults in the IGF1R-positive group.

One limitation of this study stems from its retrospective design. Further research and prospective study are necessary to fully understand the clinical implications of IGF1R expression. Furthermore, prospective clinical trials on the combination therapy of IGF1R inhibitors and ICIs are warranted. The relationship between IGF1R expression and the effects of ICI was previously unknown and requires further study. In addition, future research should investigate whether IGF1R expression is related to immune cell infiltration and elucidate the underlying mechanisms.

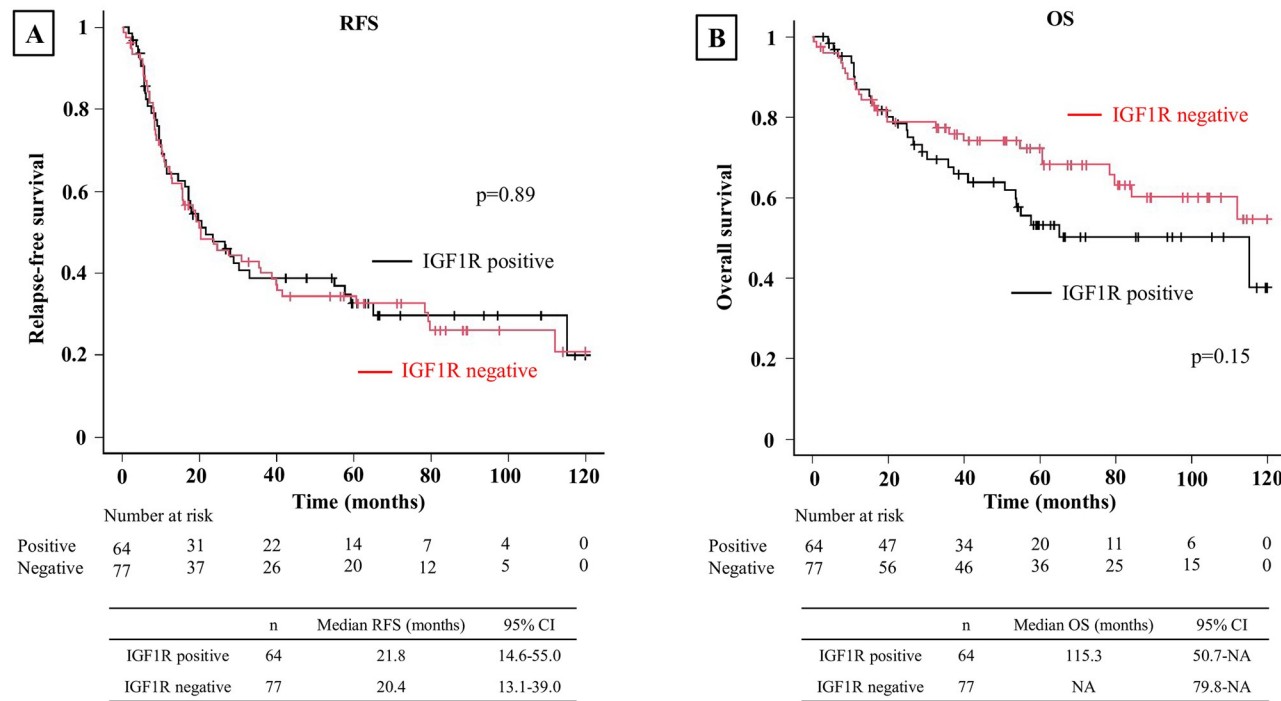

**Fig 6. Kaplan-Meier curve depicting RFS and OS in stage 3 patients.** (A) Kaplan-Meier curve depicting RFS in stage 3 patients. No significant difference in RFS was observed between the IGF1R-positive and negative groups. (B) Kaplan-Meier curve depicting OS in stage 3 patients. No significant difference was observed between the IGF1-positive and negative groups.

In conclusion, our study reveals that higher IGF1R expression correlates with poorer outcomes in terms of RFS and OS, particularly in patients with early-stage lung cancer. Furthermore, our findings indicate a relationship between IGF1R expression in lung cancer and several factors: PD-L1 expression, smoking history, and tumor size.

**Table 3. Multivariable analysis of RFS in stage 0–1 patients.**

| Factor | Reference | RFS of stage 0–1 patients (n = 494) | | | | | |
| --- | --- | --- | --- | --- | --- | --- | --- |
| | | Univariable analysis | | | Multivariable analysis | | |
| | | HR | 95%CI | p-value | HR | 95%CI | p-value |
| IGF1R | Negative | 1.71 | 1.23–2.40 | <0.01 | 1.29 | 0.85–1.96 | 0.23 |
| Age | ≤64 | 1.34 | 0.91–1.95 | 0.14 | | | |
| Sex | Female | 2.05 | 1.45–2.89 | <0.01 | 1.35 | 0.87–2.11 | 0.18 |
| Smoking history | Never | 2.41 | 1.62–3.58 | <0.01 | 1.75 | 1.04–2.95 | 0.04 |
| ECOG PS | 0 | 2.13 | 1.37–3.30 | <0.01 | 2.05 | 1.30–3.23 | <0.01 |
| Histology | NonSq | 1.52 | 1.05–2.21 | 0.03 | 0.73 | 0.45–1.18 | 0.20 |
| Adjuvant therapy | Done | 0.66 | 0.66–1.68 | 0.84 | | | |
| Pleural invasion | Negative | 1.93 | 1.35–2.76 | <0.01 | 1.32 | 0.87–2.04 | 0.19 |
| Lymphatic invasion | Negative | 2.26 | 1.57–3.25 | <0.01 | 1.70 | 1.10–2.63 | 0.02 |
| Vascular invasion | Negative | 2.68 | 1.80–3.98 | <0.01 | 1.91 | 1.17–3.12 | <0.01 |
| PD-L1 | Negative | 1.49 | 0.90–2.47 | 0.12 | | | |

RFS, relapse-free survival; HR, hazard ratio; CI, confidence intervals; IGF1R, insulin-like growth factor 1 receptor; ECOG PS, Eastern Cooperative Oncology Group Performance Status; Sq, squamous cell carcinoma; PD-L1, programmed death ligand 1.

**Table 4. Multivariable analysis of OS in stage 0–1 patients.**

| Factor | Reference | OS of stage 0–1 patients (n = 494) | | | | | |
|---|---|---|---|---|---|---|---|
| | | Univariable analysis | | | Multivariable analysis | | |
| | | HR | 95%CI | p-value | HR | 95%CI | p-value |
| IGF1R | Negative | 1.98 | 1.35–2.91 | <0.01 | 1.40 | 0.86–2.29 | 0.17 |
| Age | ≤64 | 1.26 | 0.81–1.94 | 0.31 | | | |
| Sex | Female | 2.79 | 1.82–4.29 | <0.01 | 1.69 | 0.99–2.91 | 0.06 |
| Smoking history | Never | 2.81 | 1.73–4.57 | <0.01 | 1.70 | 0.90–3.21 | 0.10 |
| ECOG PS | 0 | 2.61 | 1.60–4.28 | <0.01 | 2.24 | 1.34–3.73 | <0.01 |
| Histology | NonSq | 2.10 | 1.39–3.16 | <0.01 | 0.95 | 0.55–1.64 | 0.86 |
| Adjuvant therapy | Done | 1.14 | 0.65–1.99 | 0.65 | | | |
| Pleural invasion | Negative | 1.34 | 0.86–2.09 | 0.20 | | | |
| Lymphatic invasion | Negative | 1.39 | 0.86–2.23 | 0.18 | | | |
| Vascular invasion | Negative | 2.19 | 1.36–3.53 | <0.01 | 1.52 | 0.85–2.70 | 0.15 |
| PD-L1 | Negative | 1.71 | 0.98–3.01 | 0.06 | | | |

OS, overall survival; HR, hazard ratio; CI, confidence intervals; IGF1R, insulin-like growth factor 1 receptor; ECOG PS, Eastern Cooperative Oncology Group Performance Status; Sq, squamous cell carcinoma; PD-L1, programmed death ligand 1.

## Supporting information

**S1 Data.**
(XLSX)

## Acknowledgments

We express our gratitude to Akiko Tsuda from the Molecular Oncology and Therapeutics department at Osaka Metropolitan University Graduate School of Medicine for her valuable technical assistance.

## Author Contributions

**Conceptualization:** Hiroaki Nagamine, Masakazu Yashiro.

**Data curation:** Hiroaki Nagamine, Masakazu Yashiro, Megumi Mizutani, Akira Sugimoto, Satoshi Suzuki.

**Formal analysis:** Hiroaki Nagamine, Masakazu Yashiro, Megumi Mizutani, Akira Sugimoto, Satoshi Suzuki.

**Investigation:** Hiroaki Nagamine, Masakazu Yashiro, Megumi Mizutani, Akira Sugimoto, Satoshi Suzuki.

**Methodology:** Hiroaki Nagamine, Masakazu Yashiro, Yoshiya Matsumoto, Kenji Sawa, Hiroyasu Kaneda, Kazuhiro Yamada.

**Project administration:** Masakazu Yashiro.

**Supervision:** Masakazu Yashiro, Kazuhiro Yamada, Tetsuya Watanabe, Kazuhisa Asai.

**Validation:** Hiroaki Nagamine, Masakazu Yashiro, Megumi Mizutani, Akira Sugimoto, Kazuhiro Yamada, Satoshi Suzuki.

**Visualization:** Hiroaki Nagamine, Masakazu Yashiro.

**Writing – original draft:** Hiroaki Nagamine, Masakazu Yashiro.

**Writing – review & editing:** Hiroaki Nagamine, Masakazu Yashiro, Megumi Mizutani, Akira Sugimoto, Yoshiya Matsumoto, Yoko Tani, Kenji Sawa, Hiroyasu Kaneda, Kazuhiro Yamada, Tetsuya Watanabe, Kazuhisa Asai, Satoshi Suzuki, Tomoya Kawaguchi.

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
