## [Decision Letter · Decision Letter 0]

2 Jul 2024

PONE-D-23-43471Insulin-like Growth Factor 1 Receptor Expression Correlates with Programmed Death Ligand 1 Expression and Poor Survival in Non-small Cell Lung CancerPLOS ONE

Dear Dr. Yashiro,

Thank you for submitting your manuscript to PLOS ONE. After careful consideration, we feel that it has merit but does not fully meet PLOS ONE’s publication criteria as it currently stands. Therefore, we invite you to submit a revised version of the manuscript that addresses the points raised during the review process.

We look forward to receiving your revised manuscript.

Kind regards,

Huei-Wen Chen, Ph.D.

Academic Editor

PLOS ONE

Journal Requirements:

2. In the online submission form, you indicated that "The data underlying the results presented in the study are available from correspondence author "Masakazu Yashiro"."

**Additional Editor Comments:**

Dear Prof. Yashiro,

Many thanks for your patience. Your manuscript entitled “ Insulin-like Growth Factor 1 Receptor Expression Correlates with Programmed Death Ligand 1 Expression and Poor Survival in Non-small Cell Lung Cancer” has now been seen by 2 referees, whose comments are enclosed as below. In light of these comments, we would be interested in considering a minor revised version that addresses the comments of the referees.

The study presents compelling evidence that IGF1R expression is associated with poorer survival outcomes and higher PD-L1 expression in NSCLC patients, particularly in early stages. This highlights the potential of IGF1R as a prognostic marker and its possible role in shaping perioperative treatment strategies. Further research is needed to validate these findings, including IHC staining of the co-positive of IGF1R and PDL1? Immune cells infiltration? explore underlying mechanisms; especially, this study suggests a link between IGF1R and worse outcomes but does not delve into the underlying mechanisms. Further discussion should be added and aim to elucidate how IGF1R contributes to tumor progression, resistance to therapies, cancer stem cells, and might contribute on onco-immune regulations. and assess the value of this in early stage of targeting IGF1R in NSCLC.

Reviewers' comments:

Reviewer's Responses to Questions

**Comments to the Author**

1. Is the manuscript technically sound, and do the data support the conclusions?

Reviewer #1: Yes

Reviewer #2: Yes

2. Has the statistical analysis been performed appropriately and rigorously? 

Reviewer #1: Yes

Reviewer #2: Yes

3. Have the authors made all data underlying the findings in their manuscript fully available?

Reviewer #1: Yes

Reviewer #2: Yes

4. Is the manuscript presented in an intelligible fashion and written in standard English?

Reviewer #1: Yes

Reviewer #2: Yes

5. Review Comments to the Author

Reviewer #1: The present manuscript “Insulin-like Growth Factor 1 Receptor Expression Correlates with Programmed Death Ligand 1 Expression and Poor Survival in Non-small Cell Lung Cancer“ by Nagamine et al. describes the results of a large-scale, long-term observational study showing that IGF1R expression is associated with PD-L1 expression in the tumor and translates into worse postoperative recurrence and long-term prognosis in lung cancer.

The manuscript is well written and the methods are sound.

Combined ICI and IGF1R inhibition as possible therapeutic intervention (as mentioned by the authors) should be discussed in further detail based on current literature.

The underlying mechanistic circuitries should be also discussed in further detail.

Future correlative studies should focus on IGF1R expression and responsiveness to immune checkpoint blockade.

Reviewer #2: 1. In Keyword section IGF1R, NSCLC, OS, PD-L1, RFS should be written in full along with abbreviations for easy database search.

2. Few grammatical mistakes are there in manuscript. Please check throughout the manuscript

6. PLOS authors have the option to publish the peer review history of their article (what does this mean?). If published, this will include your full peer review and any attached files.

Reviewer #1: No

Reviewer #2: No

---

## [Author Response · Author response to Decision Letter 0]

11 Jul 2024

Editorial Office

PLOS ONE

Dear Editor-in-chief

Reference number: PONE-D-23-43471

Title: Insulin-like Growth Factor 1 Receptor Expression Correlates with Programmed Death Ligand 1 Expression and Poor Survival in Non-small Cell Lung Cancer

Authors：Hiroaki Nagamine, et al.

We thank you and the reviewers for your thoughtful suggestions and insights. The manuscript has benefited from these insightful suggestions. I look forward to working with you and the reviewers to move this manuscript closer to publication in the PLOS ONE.

The manuscript has been rechecked and the necessary changes have been made in accordance with the reviewers’ suggestions. The responses to all comments have been prepared and attached herewith. 

We have added a discussion based on the editor and reviewers’ comments. As you pointed out, clarifying whether IGF1R expression is related to immune cell infiltration is an important future task, and we have added this point to the limitations section. In addition, as it would be beneficial, we have also added HE-stained images to Figure 1. In addition, some RFS were mistakenly listed as PFS, which has been corrected (p14, line 174 and 176). The changes are highlighted in yellow.

All authors are aware of the content of this manuscript.

Thank you for your consideration. I look forward to hearing from you.

Sincerely,

Masakazu Yashiro

Molecular Oncology and Therapeutics, Osaka Metropolitan University, Graduate School of Medicine, 1-4-3 Asahimachi, Abeno-ku, Osaka 545-8585, Japan

Email: i21496f@omu.ac.jp

Reply to Reviewer 1

Combined ICI and IGF1R inhibition as possible therapeutic intervention (as mentioned by the authors) should be discussed in further detail based on current literature.

⇒

[Response]

Thank you for pointing out this critical point.

Although clinical trials have not yet been conducted, several studies in mouse models have reported that combining an IGF1R inhibitor and immune checkpoint inhibitors (ICIs), such as PD-1 inhibitors, synergistically enhances antitumor effects. IGF1R influences the tumor microenvironment in various ways, and the inhibition of IGF1R is known to affect the activation of cytotoxic CD8+ T cells, which are important for the efficacy of ICIs. Therefore, combining IGF1R inhibitors with ICIs may enhance antitumor effects. This notion has been supported by recent studies, which we have referenced accordingly.

・However, recent mouse experiments have revealed that reduced IGF1R expression enhances the effects of ICI [23], and that IGF1R expression correlates with the tumor microenvironment [24], including immune cells. Combining IGF1R inhibitors with ICI may hold promise for future therapies, especially as perioperative treatment for early-stage lung cancer. 

↓

・However, recent experiments in mice have demonstrated that the combination of an IGF1R inhibitor and an anti-PD-1 antibody synergistically inhibits tumor growth [32, 33]. IGF1R expression is correlated with the tumor microenvironment and inhibits the activation of effector cytotoxic CD8+ T cells [34, 35]. Therefore, inhibition of the IGF1R pathway has the potential to enhance the effectiveness of ICI by activating effector cytotoxic CD8 T cells [36]. From those results, Combining IGF1R inhibitors with ICI may hold promise for future therapies, especially as perioperative treatment for early-stage lung cancer. 

The underlying mechanistic circuitries should be also discussed in further detail.

⇒

[Response]

Thank you for pointing out the important details. We have added discussions on the underlying mechanisms, citing various studies. Specifically, we have added discussions on how IGF1R is involved in cell proliferation, the high recurrence rates in early-stage lung cancer, and how IGF1R expression is related to chemoresistance.

・The IGF1R-positive group also showed significantly larger tumor sizes. IGF1R is a factor associated with tumor growth [4], consistent with a relationship between IGF1R expression and tumor size. In addition, IGF1R expression was associated with shorter RFS and OS, particularly in early-stage lung cancer.

↓

・The IGF1R-positive group also showed significantly larger tumor sizes. IGF1R is a factor associated with tumor growth [4]. Upon binding with its ligands IGF1 or IGF2, IGF1R activates downstream signaling pathways such as the phosphoinositide 3-kinase (PI3K)/Akt and extracellular signal-regulated kinase (ERK) pathways, which leads to tumor cell proliferation [18-20]. This is consistent with the result of the relationship between IGF1R expression and tumor size. In addition, IGF1R expression was associated with shorter RFS and OS, particularly in early-stage lung cancer. It has been known that stage 1 has a lower frequency and amount of postoperative molecular residual disease (MRD) compared to stage 2 or 3 in lung cancer [21]. Although the immune system may eliminate minimal MRD in early-stage lung cancer, our results suggested that IGF1R expression was associated with PD-L1 expression, which might have allowed cancer cells to escape immune surveillance [22, 23] and increased the likelihood of relapse in IGF1R-positive patients. Additionally, IGF1R expression may facilitate tumor growth, leading to earlier recurrence compared to IGF1R-negative cases. Also, IGF1R expression has been known to correlate with the expression of ATP–binding cassette subfamily G member 2 (ABCG2) [24], a potential marker for cancer stem cells [25] and involved in chemoresistance [26]. These findings might explain the treatment resistance of recurrent tumors and may account for the worse OS observed in early-stage lung cancer with IGF1R positivity, where recurrence is more frequent.

Future correlative studies should focus on IGF1R expression and responsiveness to immune checkpoint blockade.

⇒

[Response]

Thank you for pointing out these important details. There are currently no studies evaluating the relationship between IGF1R expression and the effects of ICI. This is a significant issue for the future. Therefore, based on your suggestions, we have added the following text as a limitation.

・Furthermore, prospective clinical trials on the combination therapy of IGF1R inhibitors and ICIs are warranted. The relationship between IGF1R expression and the effects of ICI was previously unknown and requires further study. In addition, future research should investigate whether IGF1R expression is related to immune cell infiltration and elucidate the underlying mechanisms. 

Reply to Reviewer 2

1. In Keyword section IGF1R, NSCLC, OS, PD-L1, RFS should be written in full along with abbreviations for easy database search.

⇒

[Response]

Thank you for your comment. I have added the full notation of each abbreviation in the keywords section.

・Keywords: IGF1R, lung cancer, NSCLC, OS, PD-L1, RFS

↓

・Keywords: IGF1R, insulin-like growth factor 1 receptor, lung cancer, NSCLC, non-small cell lung cancer, OS, overall survival, PD-L1, programmed cell death ligand 1, RFS, recurrence free survival

2. Few grammatical mistakes are there in manuscript. Please check throughout the manuscript.

⇒

[Response]

Thank you for pointing this out. We have reviewed the text again and made some changes.　Specifically, they are as follows.

p11, line 63

・Moreover, smoking is known to be involved in tumor-induced IGF1R expression [10].

↓

・Moreover, smoking has been known to be involved in tumor-induced IGF1R expression [10].

p13, line 87

・This study was approved by the Osaka City university Ethics Committee (reference number 2019-006).

↓

・This study was approved by the Osaka City University Ethics Committee (reference number 2019-006).

p13, line 89

・This study has been conducted according to the principles of the declaration of Helsinki.

↓

・This study was conducted according to the principles of the Declaration of Helsinki.

p15, line 111 to 115

・Multiple logistic regression analysis was utilized to analyze the relationship between IGF1R and PD-L1 expression, incorporating factors age, sex, smoking history, Eastern Cooperative Oncology Group Performance Status (ECOG PS), histology, pStage, pleural invasion, lymphatic invasion, and vascular invasion.

↓

Multiple logistic regression analysis was utilized to analyze the relationship between IGF1R and PD-L1 expression, incorporating age, sex, smoking history, Eastern Cooperative Oncology Group Performance Status (ECOG PS), histology, pStage, pleural invasion, lymphatic invasion, and vascular invasion as factors.

p15, line 115 to 117

・The Mann–Whitney U test was applied to evaluate differences in smoking index or maximum tumor diameter between the IGF1R-positive and negative groups.

↓

・The Mann–Whitney U test evaluated differences in smoking index or maximum tumor diameter between the IGF1R-positive and negative groups.

p25, line 218 to 219

・However, due to the favorable prognosis of early-stage lung cancer, it was less amenable to the benefits of neoadjuvant and adjuvant chemotherapy.

↓

・However, due to the favorable prognosis of early-stage lung cancer, it is less amenable to the benefits of neoadjuvant and adjuvant chemotherapy.

---

## [Editor Report · Decision Letter 1]

23 Jul 2024

Insulin-like Growth Factor 1 Receptor Expression Correlates with Programmed Death Ligand 1 Expression and Poor Survival in Non-small Cell Lung Cancer

PONE-D-23-43471R1

Dear Dr. Yashiro,

We’re pleased to inform you that your manuscript has been judged scientifically suitable for publication and will be formally accepted for publication once it meets all outstanding technical requirements.

Kind regards,

Huei-Wen Chen, Ph.D.

Academic Editor

PLOS ONE

---

## [Editor Report · Acceptance letter]

26 Jul 2024

PONE-D-23-43471R1 

PLOS ONE

Dear Dr. Yashiro, 

I'm pleased to inform you that your manuscript has been deemed suitable for publication in PLOS ONE. Congratulations! Your manuscript is now being handed over to our production team.

Kind regards, 

on behalf of

Prof. Huei-Wen Chen 

Academic Editor

PLOS ONE